# An Analysis of the Vibration Transmission Properties of Assemblies Using Honeycomb Paperboard and Expanded Polyethylene

**DOI:** 10.3390/ma16196554

**Published:** 2023-10-04

**Authors:** Yueqing Xing, Deqiang Sun, Zelong Deng

**Affiliations:** College of Bioresources Chemical and Materials Engineering (College of Flexible Electronics), Shaanxi University of Science & Technology, Xi’an 710021, China

**Keywords:** assembly, vibration transmission rate, damping energy dissipation, honeycomb paperboard

## Abstract

In the process of logistics, shock and vibration are the most important factors contributing to product damage. Assembling honeycomb paperboard and EPE is commonly used to provide cushioning and anti-vibration effects to materials. Therefore, it is necessary to study the vibration transmission properties of this kind of assembly in the anti-vibration process. The aim of this paper was to experimentally determine the vibration transmission properties of assemblies with honeycomb paperboard and expanded polyethylene (EPE). Through a sinusoidal sweep vibration test of this assembly, the vibration transmission characteristic curves of assemblies with honeycomb paperboard and EPE of different thicknesses were obtained and compared. Assuming the assembly and mass block as a single degree of freedom with a small damping linear system, the damping energy dissipation of the assembly and the resonance frequency were obtained. The vibration transmission property curves of the assembly can be divided into four regions. With an excitation acceleration of 0.5 g and a honeycomb paperboard with a thickness of 60 mm (F60), the vibration transmission rate and the resonance frequency—of the material dampened with EPE at a thickness of 60 mm (E60), and the assembly (F30/E30) with a 30 mm thick honeycomb paperboard and 30 mm thick EPE—increased by −2.5% and −17.5%, −86.9% and 79.3%, and −95.9% and −85.7%, respectively. Compared to the assembly with 20 mm thick honeycomb and 20 mm thick EPE (F20/E20), the vibration transmission rate, the resonance frequency, and the material damping and damping energy dissipation of F40/E20, F30/E30, and F20/E40 increased by 75.6%, 48.3%, and 66.1%; 1.2%, −21.5%, and −38.9%; 241.5%, 82.8%, and 13.3%; and 12.5%, 98.9%, and 106.8%, respectively. Compared to F60 and E60, the damping energy dissipation of F30/E30 increased by 2816.7% and 133.3%, respectively. The assembly of F30/E30 has the smallest vibration transmission rate and the most vibration energy dissipation among these assemblies. This means that the assembly of F30/E30 absorbs the most external vibration energy, while the acceleration that is transmitted to the internal product is minimal. Therefore, in the design of cushioning packaging, according to the characteristics and natural frequency of the internal products, an appropriate assembly can be selected, which should have a lower vibration transmission rate and more vibration energy dissipation, and should not resonate with the internal product. This will provide a theoretical basis for the design of cushioning packaging.

## 1. Introduction

A packaging system is generally composed of complex internal products, non-linear viscoelastic cushioning materials, and outer packaging, such as EPE and a honeycomb box. Due to the elasticity and damping ability of the cushioning material, on the one hand, the response of the product to external vibration excitation is reduced, and the plastic deformation of the cushioning material occurs, dissipating the energy of the external vibration excitation [1]. Therefore, under normal circumstances, the damage effect of vibration on the packaging system and products is relatively small. However, when the natural vibration frequency of the packaging system is equal to or close to the natural vibration frequency of the product or the key parts of the product, there is a possibility of resonance, and the exciting force will be amplified, which will have a certain destructive effect on the packaging product. The conductive action of cushioning packaging materials against external vibration forces can be quantitatively described using a vibration transmission curve. In the design of anti-vibration packaging, based on the vibration transmission curve of the cushioning material, the frequency of the packaging system is adjusted away from the natural frequency of the product to reduce the dynamic stress and extend the service life of the product. Firstly, the excitation and response signals of the system were analyzed and the relationship between the system response and excitation is determined at each excitation frequency. The corresponding relationship is the vibration transmission rate. The *T*_r_-*f* curve was drawn with the excitation frequency as the horizontal coordinate and the vibration transmission rate as the vertical coordinate [2].

Honeycomb paperboard is a kind of nonmetallic, composite, porous, and solid sandwich structure, which has good cushioning and anti-vibration performance and is widely used in the fields of product protection and packaging engineering [3,4,5,6]. Honeycomb paperboard is an environmentally friendly cushioning packaging material with a novel structure, light weight, high specific stiffness, and high specific strength [7]. In the dynamic protection design of an actual packaging system, researchers have conducted many studies on the cushioning performance of honeycomb paperboard, including its static and dynamic compression deformation mechanism, its cushioning energy absorption characteristics, and the construction of the constitutive model [8,9,10,11].

EPE has good cushioning and sound insulation properties, and is often used in protective and shock-energy-absorbing materials for packaging. In terms of its vibration transmission properties, EPE mainly has vibration energy absorption, vibration isolation, frequency response, and other properties [12]. EPE is a low-density, semi-rigid, closed cell structure of plastic foam, with a light weight, soft surface, and resistance to multiple impacts. It is a good packaging material with good cushioning performance, widely used in packaging, construction, electronics, and other fields [13,14,15,16]. In packaging applications, a reasonable selection of the thickness, density, and structural shape of EPE, alongside a combination of other packaging materials such as honeycomb paperboard, can enhance its shock absorption and anti-vibration effect.

Guo Yanfeng [17,18] has already studied the buffering property and vibration transmission characteristics of four kinds of honeycomb paperboards with different thicknesses (20, 30, 40, and 50 mm) through experiments. Zhang Junling [19] drew vibration transmission curves by conducting theoretical and experimental studies on the vibration resistance and buffering properties of honeycomb paperboards with different thicknesses (20, 30, 40, and 50 mm). J. Park [20] compared the vibration transmission rate and resonance frequency of corrugated cardboard with different corrugated shapes under different static stresses by using a sinusoidal sweep frequency vibration test, and evaluated the damping ratio and maximum dynamic stress using linear vibration theory. Yueqing Xing [21] analyzed the excitation and response signals of a processing system, determined the relationship between the response and excitation at each excitation frequency, namely the transmissibility, and then drew a vibration transmissibility curve. Liang Ning [22] calculated the damping energy dissipation of the resonance vibration point of a honeycomb paperboard mass system. If the vibration transmission property curve of a packaging material is obtained, the *W*_c_ of the packaging system can be calculated at any frequency. With an increase in honeycomb structure parameters, the damping ability of honeycomb board decreases, and the damping energy dissipation of the resonance vibration point increases. With the increase in load mass, the damping energy dissipation of the system resonance point also increases. Gong Guifeng [23] conducted a study on the vibration transmission characteristics of a cylindrical air gasket. Through a sinusoidal sweep vibration test, they tested the cylindrical air gasket and the vibration system, subject to the determined load mass, within the frequency range of 3–100 Hz, so as to obtain a vibration transmission curve. Yueqing Xing [24] studied the static crushing responses of EPP foam and found that EPP foam’s density had a significantly greater influence on static compressive performance than the foam’s thickness. Yueqing Xing [25] studied the dynamic crushing behavior of EVA Copolymer Foam based on the energy method.

Many scholars have conducted a lot of research on the damping performance and *W_c_* of composite sandwich structures, especially the damping performance of composite materials. Wang [26,27] studied vibration analysis methods for complex transportation packaging systems. Matte [28] proposed a hybrid identification method of simulation and experimentation to evaluate the elastic and damping properties of honeycomb sandwich structures with low plane/core stiffness ratios. Tsai [29] studied the damping response of epoxy-based nanocomposites and composite sandwich structures with nanocomposites as core materials. The damping performance of nanocomposites and sandwich structures was measured via the half-power method and forced vibration method, and the resonance frequency obtained via a vibration experiment was used to evaluate the flexural stiffness of the material system. Wei [30] investigated resonance techniques for evaluating damping properties and stiffness coefficients of composite structures, and the results show that the resonance frequency (natural frequency) of a material or system is a function of its elastic properties, size, and mass.

The above studies are either analyses of the vibration transmission rate of the honeycomb paperboard or other single cushioning materials, or studies of the influence of external conditions on the vibration transmission of cushioning materials.

Because honeycomb paperboard and EPE have certain differences in vibration resistance at different frequencies, the assembly of honeycomb paperboard and EPE with different thickness changes the vibration resistance in a specific frequency band, and combines the actual transportation environment and the different natural frequencies of the internal products to prevent the occurrence of resonance phenomena. At the same time, the vibration transmission rate of the assembly is reduced to the greatest extent, and the damping energy dissipation of the assembly needs to be as high as possible. Therefore, we need to systematically and comprehensively study the vibration transmission property of the assemblies to provide a better theoretical basis for cushioning packaging design.

## 2. Materials and Methods

Prepare honeycomb paperboard and EPE specimens of different thickness, glue the two kinds of materials together with hot melt adhesive to form the different assemblies, then calibrate the test equipment, and carry out sine sweep frequency vibration test in strict accordance with the vibration experiment standards of cushioning materials. The test scheme and flow chart are shown in Figure 1.

### 2.1. Materials

The EPE foams were acquired from Suzhou Shunsheng Packaging Material Co., Ltd., Suzhou, China, with density of 15.3 kg/m^3^. The honeycomb paperboard was acquired from Xi′an Haihong Packaging Co., Ltd., Xi′an, China, with an aperture of 6 mm.

Specimens have a cross-section of 100 mm × 100 mm. Specimen labels include a letter and 1 set of numbers; for example, F and E represent the uniform code of the honeycomb paperboard specimen and EPE specimen, respectively. The following two-digit number is a thickness of the specimen. Thus, specimen F20/E40 is the assembly specimen with a thickness of 20 mm for honeycomb paperboard and a thickness of 40 mm for EPE. The honeycomb paperboard and EPE were bonded to form an assembly using hot-melt adhesive.

The test specimen parameters are shown in Table 1.

### 2.2. Vibration Test Equipment and Method

In order to obtain the vibration transmission curve of cushioning material, it is necessary to carry out sinusoidal sweeping vibration test on the cushioning packaging system composed of cushioning material and mass block. Sinusoidal sweep vibration makes the frequency change linearly or exponentially with time [31], and has the characteristic of energy concentration, flexible sweep frequency range and high signal-to-noise ratio. The sinusoidal sweep frequency vibration test method is in strict accordance with GB8169-2008 [32]. The test equipment is electric vibration table of DC-300-2, made by Suzhou Test Instrument Co., Ltd., Suzhou, China. The vibration test specimens are prepared for more than 24 h in an environment of 20 ± 2 °C and relative humidity of 45%.

There are two acceleration sensors used in the test. One is installed on the mass block to monitor the output signal with a sensitivity of 24.50 V/g, and the other is installed on the DC-300-2 electric vibration table to control the input signal with a sensitivity of 24.40 V/g. The test parameters are shown in Table 2.

### 2.3. Vibration Test System

The vibration test system is composed of cushioning material, mass block, charge amplifier, 16-channel A/D conversion board, acceleration sensor, computer, vibration controller RC-2000 and vibration table of DC-300-2. Figure 2a shows the actual vibration table control system and data acquisition and analysis system, and Figure 2b shows the vibration transmission property test system. The vibration test principle is shown in Figure 3. The vibration table of DC-300-2 and sinusoidal vibration controller of RC-2000 are excitation devices of the test system. The vibration controller performs sinusoidal sweeping frequency excitation on the cushioning materials and mass blocks fixed on the vibration table in logarithmic way from 3 to 150 Hz, and increases the frequency through the resonance point from 3 Hz. At this time, the vibration transmission rate reaches the maximum value, and then the vibration transmission gradually decreases to about 0.2. Two acceleration sensors are fixed on the vibration table and the cushioning material to collect the excitation signal and the response signal, respectively, and convert the obtained physical signal into electrical signal. The charge amplifier of 1 and 2 amplify the two electrical signals, and the 16-channel A/D converter converts the electrical signal into digital signal. The computer reads the excitation and response digital signals and performs signal acquisition, smoothing and filtering, time and frequency domain analysis and processing, and outputs the *T_r_–f* curve.

### 2.4. Vibration Property Criteria

The system composed of an assembly and mass block is regarded as a linear system with single degree of freedom and small damping. As shown in Figure 4, *m* is the mass of the assembly and mass block, *y*(t) is the external excitation displacement, *x*(t) is the response displacement of the assembly, *k* and *c* are the equivalent stiffness coefficient and damping coefficient of the packaging material, respectively.

The differential equation of the system under vibration is as Equation (1) [22]:(1)$$m\ddot{x}=-k(x-y)-c(\ddot{x}-\ddot{y})$$

The complex displacement excitation received by the assembly during transportation can be simplified into a linear superposition of multiple harmonic excitation, so it can be expressed with Equation (2):*y* = *A*sin*ωt*(2)
where *A* is the amplitude of the harmonic excitation displacement, *ω* is the excitation frequency, and *t* is the excitation duration.

By substituting Equation (2) into Equation (1), dividing both left and right ends by *m*, and introducing damping ratio *ξ*, natural frequency *ω*_n_, and frequency ratio *λ* can be expressed with Equation (3):(3)$$\xi =\frac{c}{2m\omega_{n}}\;\;\;\;\;\;\omega_{n}=\frac{k}{m}\;\lambda =\frac{\omega }{\omega_{n}}$$

Under the vibration condition of simple harmonic excitation, the material damping *c* can be expressed with Equation (4) [14]:*C* = 2*mξω_n_* = 4*πmξf_n_*(4)

Under sinusoidal excitation, the formula for calculating the vibration transmission rate *T_r_* of the single-degree-of-freedom system composed of mass blocks and assembly is as Equation (5) [22]:(5)$$T_{r}=\sqrt{\frac{1+{\left(2\xi \lambda \right)}^{2}}{{\left(1-\lambda^{2}\right)}^{2}+{\left(2\xi \lambda \right)}^{2}}}$$

When the system resonates, the frequency ratio is λ ≈1, and the formula for calculating the material damping ratio *ξ* can be obtained from Equations (5) and (6):(6)$$\xi =\frac{1}{2}\sqrt{\frac{1}{T_{r}^{2}-1}}$$

*W*_c_ refers to the conversion of system kinetic energy into heat or other forms of energy loss by increasing system damping, so as to achieve the purpose of shock absorption, vibration suppression, or energy consumption [34]. System damping energy dissipation can effectively reduce the vibration and resonance phenomena of a structure, thereby reducing the risk of fatigue damage and destruction of the protected product and increasing the service life and reliability of packaging materials [35].

When the assembly vibrates under a certain mass block, with a certain frequency point as the period, the energy dissipation is the mechanical energy consumed by the cushioning material damping in a period. By integrating the work carried out by the damping force in a period, *W_c_* can be expressed with Equation (7) [22]:(7)$$W_{c}=\oint c\ddot{x}\left(t\right)dx=\int_{0}^{T}c\dot{x}(t)dx$$

The steady-state response expression of a linear system with single degree of freedom under harmonic excitation acceleration is as Equation (8):(8)$$\ddot{x}\left(t\right)=X_{2}\sin\left(\omega t-\beta \right)$$

By integrating Equation (8), the velocity expression can be obtained with Equation (9):(9)$$\dot{x}\left(t\right)=\frac{-X}{\omega }\cos\left(\omega t-\beta \right)$$

By substituting Equation (9) into Equation (7) for integral operation, *W_c_* in a period of a single degree linear system at any frequency point under harmonic excitation acceleration can be expressed with Equation (10) [22]:(10)$$W_{c}=\frac{cX_{2}^{2}}{\omega^{2}}\int_{0}^{\frac{2\pi }{\omega }}\cos\left(\omega t-\beta \right)dt=\frac{cX_{2}^{2}}{8\pi^{2}f^{3}}=\frac{cB^{2}T_{r}^{2}}{8\pi^{2}f^{3}}$$

In Equation (10), *B* is the amplitude of excitation acceleration, which is 0.5 g, that is, 4.9 m/s^2^. Equation (10) shows that *W_c_* is related to the vibration transmission rate *T_r_* and natural frequency of the material, and the vibration transmission curve can be obtained to calculate the damping energy dissipation of the packaging system.

## 3. Results

The typical vibration transmission curves of honeycomb paperboard and assembly are shown in Figure 5 and Figure 6, respectively. The trends of these two curves is very similar. The typical vibration transmission curve of the assembly is divided into four stages, which is similar to that of the honeycomb paperboard [36]: the frequency of plateau region is about 0–10 Hz, and the vibration transmission rate always fluctuates up and down nearby, and the fluctuation range is not large. The results show that the vibration is transmitted by a law equal to 1 to 1, and the maximum acceleration transmitted to the product is small. In the amplification region in Figure 6, which is in the range of resonance frequency of about 20 Hz, the vibration transmission rate rapidly increases to form a peak value, and then rapidly decreases to about 1. In this frequency range, the curve basically takes the resonance frequency as the center line and is basically symmetrical. The frequency of fluctuation region in Figure 5 is about 25–50 Hz, and 1–2 peaks will be formed on the curve, but the peaks are far smaller than the maximum peak. The frequency of the attenuation region in Figure 6 is about 50–150 Hz. The curve basically shows a trend of attenuation; at this stage, the vibration transmitted to the product from the environment is reduced, and the packaging system has a vibration reduction function.

The reason for this trend in the typical vibration transmission property curves in Figure 5 and Figure 6 is that in the low-frequency segment, the excitation frequency is far less than the natural frequency of the material system, the anti-vibration performance of the material is not reflected, and the excitation acceleration is basically transmitted to the product at 1 to 1. With the sweep frequency test, the excitation frequency gradually increases; when it is close to the natural frequency of the system, resonance phenomena occurs, and the acceleration transmitted to the system will be amplified. At this time, the vibration of the environment (force, displacement, or acceleration) will be multiplied, tens of times or hundreds of times amplified and transmitted to the product, which is very likely to lead to damage to the product. When the frequency continues to increase, the honeycomb paperboard will show the effect of cushioning and anti-vibration, so that the acceleration transmitted to the product is partially blocked, so as to protect the product and prevent its being damaged.

### 3.1. Response Acceleration and Vibration Transmission Property Curve

Sinusoidal sweep vibration tests were carried out on the honeycomb paperboard, EPE, and assembly with a thickness of 60 mm, respectively. The response acceleration–frequency curves and vibration transmission rate–frequency curves were obtained, as shown in Figure 7a,b.

It can be seen from Figure 7a,b that when a thickness of the five specimens is the same, the resonance frequency of the honeycomb paperboard and EPE is 93 Hz and 13 Hz, respectively. The resonance frequency of the assembly is between 13 Hz and 93 Hz. With the increase in a thickness of EPE and the decrease in a thickness of honeycomb paperboard, the resonance frequency of the assembly decreases obviously, and the frequency interval of the amplification region of assemblies is obviously smaller than that of honeycomb paperboard. Among the five kinds of test specimens, when the excitation acceleration is 0.5 g and the assembly of F30/E30 has the same thickness of honeycomb paperboard and EPE, the response acceleration and vibration transmission rate of the resonance point are the smallest, indicating that the assembly of F30/E30 has the best protection ability when it acts as the cushioning structure in the circulation environment with a frequency of 3–150 Hz. The acceleration transmitted to the product by the cushioning material is minimal. 

Figure 8a,b shows the response–acceleration curve and vibration–transmission property curve of the assemblies for F30/E10, F30/E20, F30/E30 and F30/E40. As can be seen from Figure 8a,b, the vibration transmission rate does not increase in a linear manner, as with the increasing thickness of the assembly. With the increase in thickness for EPE, the corresponding maximum response acceleration and vibration transmission rates increase. The vibration transmission rate of the assembly of F30/E30 is the smallest and the maximum vibration transmission rate is less than 7, and the protective ability for the product is the best among the four specimens. The fluctuation region of the assembly of F30/E30 is more obvious than others. In addition to the maximum peak response acceleration of the resonance frequency point, there are several small peaks, and other assemblies do not have obvious fluctuation regions. The resonance frequency range of the four assemblies is 15–35 Hz. When this kind of assembly is used as a cushioning material for packaging design, the appropriate frequency of the assembly can be selected according to the natural frequency and characteristics of the protected products, to avoid the occurrence of resonance and damage to the internal products.

Figure 9a,b shows the response acceleration curve and vibration transmission rate curve of the assemblies of F20/E20, F30/E20 and F40/E20. As can be seen from Figure 9a,b, the vibration transmission rate for the assembly of F20/E20 is the smallest and the maximum vibration transmission rate value is about 5, and the protective ability for the protected product is the best among these assemblies. The vibration transmission rate of the other two assemblies is relatively close, reaching more than 8. The resonance frequency range of the three assemblies is 20–25 Hz. Compared to F20/E20, the vibration transmission rate of F30/E20 and F40/E20 increased by 89.4% and 78.7%, respectively.

### 3.2. Analysis of Damping Performance of Assemblies

When the structural parameters, thickness, and mass of the tested material were determined, the resonance frequency and damping ratio of the system were fixed. Therefore, the damping, damping ratio, and *W_c_* of the assembly can be calculated via Equation (4), Equation (6), and Equation (10), respectively. The results are shown in Table 1. The variations of the resonance frequency and damping of the assemblies are shown in Figure 9a,b.

As shown in Table 3 and Figure 10a,b, at the same excitation acceleration, compared to E60, the resonance frequency *f*_n_, maximum vibration transmission rate *T*_m_, material damping *c* of F60 increased by 660.9%, 24.4%, 2348.3%, respectively. Therefore, under the same thickness, the damping, the vibration transmission rate, and the resonance frequency of honeycomb paperboard are larger than those of EPE.

Compared to the assembly of F60, the resonance frequency *f_n_*, maximum vibration transmission rate *T*_m_, material damping *c* of the assembly of F40/E20 increased by −73.3%, 5.1%, −73.3%, respectively. For the assembly of F20/E40, the resonance frequency *f_n_*, maximum vibration transmission rate *T*_m_, material damping *c* increased by −83.9%, −0.6%,−91.1%, respectively, compared to the assembly of F60. Compared to the assembly of F20/E20, the resonance frequency *f*_n_, maximum vibration transmission rate *T*_m_, material damping *c* of the assembly of F40/E20 increased by 1.2%, 75.6%, 240.7%, respectively. In the assemblies of E60, F40/E20, F30/E30, F20/E40, and F60, the damping of the assemblies increased obviously with a thickness of honeycomb paperboard increases and a thickness of EPE decreases. The damping performance of honeycomb paperboard was better than EPE, and the thickness was an important factor affecting the damping performance of the assemblies.

### 3.3. Damping Energy Dissipation

System resonance means that under a certain frequency, the vibration amplitude generated by the system after being stimulated by the outside world becomes large, and the system absorbs the most external energy during the resonance frequency. The resonance point is the point at which the system has the largest amplitude at a certain frequency. However, resonance can also lead to the loss of control and destruction of the system, so measures need to be taken to reduce the adverse effects caused by resonance. The goal of damping energy analysis is to evaluate the damping performance of the system near the resonance point to ensure that the system can effectively dissipate energy during vibration, reduce amplitude, and reduce the impact of resonance.

It can be seen from Table 3 that the damping energy dissipation of the honeycomb paperboard with a thickness of 60 mm and EPE with a thickness of 60 mm at the resonance point is much smaller than that of assembly with a thickness of 60 mm, indicating that the energy absorbed by the assembly at the resonance frequency is greater than that of the single-honeycomb paperboard and single EPE. At the same time, as a thickness of EPE increased from 20 mm to 40 mm, the damping energy dissipation of the resonance point also increased, and the increasing trend is almost a quadratic function, which indicates that the damping energy dissipation of the assembly increased with the increase in EPE thickness in the assembly of the same thickness. Compared to EPE with a thickness of 60 mm, the *W_c_* of honeycomb paperboard with a thickness of 60 mm increased by −0.92%. Compared to the assembly of F20/E40, the damping energy dissipation of the assembly of F40/E20 increased by 34.3%.

It can be seen that the damping energy dissipation at the resonant point of the assembly increases with the increasing thickness of the honeycomb paperboard and EPE. The damping energy dissipation of the amplifier region and the resonant point of the honeycomb paperboard also increased with the increasing thickness of the paperboard, while the damping energy dissipation of the plateau region of the honeycomb paperboard decreased with the increasing thickness of the paperboard. The influence of thickness on damping energy dissipation in the attenuation region of the honeycomb paperboard is not obvious [22].

## 4. Discussion

It can be seen that the damping and resonance frequency of the assembly are between those of honeycomb paperboard and EPE, but the damping energy dissipation of the resonant vibration point of the assembly is far greater than that of honeycomb paperboard and EPE, indicating that the combination of honeycomb paperboard and EPE as a cushioning packaging material can increase the absorption of vibration energy and reduce the vibration energy transferred to the internal products.

The damping and resonance frequency of honeycomb paperboard are much larger than that of EPE with the same thickness, but the *W_c_* of EPE is larger than that of honeycomb paperboard with the same thickness. The results show that EPE can absorb more vibration energy than honeycomb under the same thickness condition. When a thickness of honeycomb paperboard and of the EPE in the assembly are equal, the vibration transmission rate will be lower. Therefore, in the design of cushioning packaging, according to the characteristics and natural frequency of the contained products, the appropriate assembly can be selected for packaging, which should have a lower vibration transmission rate, more vibration energy dissipation, and which will not resonate with the contained product.

## 5. Conclusions

Based on the sinusoidal sweep vibration test, the response acceleration, vibration transmission rate, resonance frequency, damping, and *W_c_* have been obtained. The following conclusions have been drawn:

(1)The obtained vibration transmission rate curve shows that the assembly generally has four stages, according to the frequency range. In the amplification region, the acceleration transmitted to the product will be multiplied by several times, especially at the resonance frequency, and the product is likely to be damaged. In the attenuation region, the protective ability for the product is better, and the response acceleration to the product is small.(2)When the aperture of honeycomb paperboard and the density of EPE were constant, the vibration transmission rate and damping energy dissipation of different assemblies were studied. Among all the assemblies, the assembly of F30/E30 had better damping energy dissipation and a lower vibration transmission rate. In the cushioning packaging design, it can be assumed that a thickness of the honeycomb paperboard and EPE are equal; then, the assembly has a lower vibration transmission rate and greater damping energy dissipation, which can minimize the maximum acceleration value transmitted to the internal product and protect the product from being damaged.

The aim of this study was to examine the influence of thickness on the vibration transmission rate and the frequency and damping energy dissipation of the assembly when the structural parameters of EPE and honeycomb paperboard are fixed. By changing the thickness to adjust the frequency of the assembly, it is possible to prevent resonance with the internal product, and to reduce the vibration transmission rate, increase the damping energy dissipation of the assembly. The results of this study can provide a theoretical basis for the selection of the thickness of honeycomb paperboard and EPE in the assembly design of cushioning packaging.

The structural parameters of honeycomb paperboard and EPE also affect the vibration transmission property of the assembly, such as the cell side length of honeycomb paperboard and the density of EPE. The influence of these structural parameters on the vibration transmission property of the assembly will be studied later.

## Figures and Tables

**Figure 1 materials-16-06554-f001:**
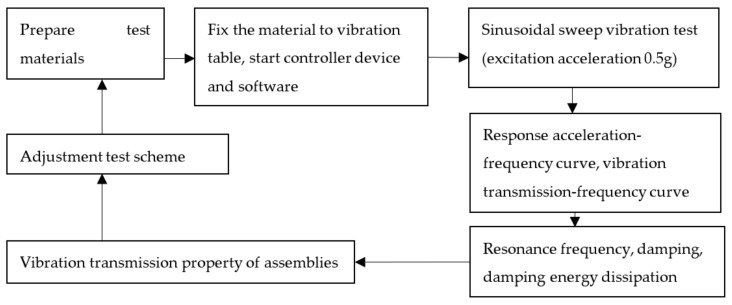
The test scheme.

**Figure 2 materials-16-06554-f002:**
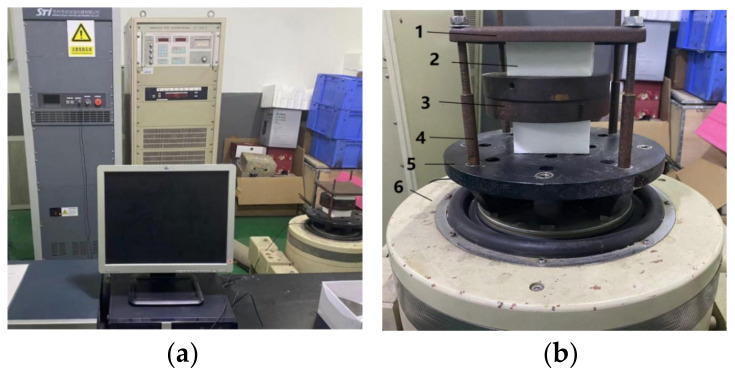
Vibration transmission property test system. (**a**) The actual vibration table control system; (**b**) the vibration table and specimens. 1. Cover plate; 2. test specimen; 3. mass block; 4. screw arbor; 5. installation table; 6. vibration table.

**Figure 3 materials-16-06554-f003:**
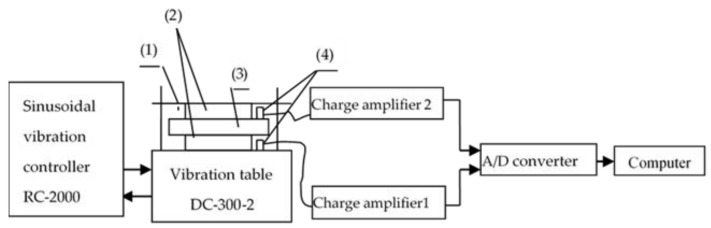
Test system of vibration transmissibility of cushioning material. 1. Clamping device; 2. cushioning material; 3. mass block; 4. acceleration sensor.

**Figure 4 materials-16-06554-f004:**
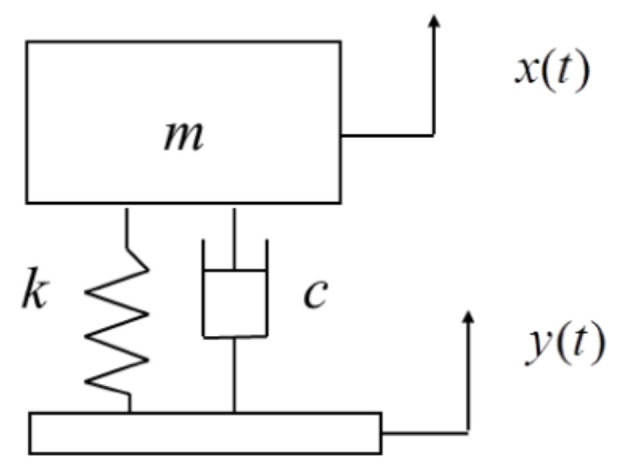
Vibration mechanics model of a linear system with single degrees of freedom under the excitation of a simple harmonic support [33].

**Figure 5 materials-16-06554-f005:**
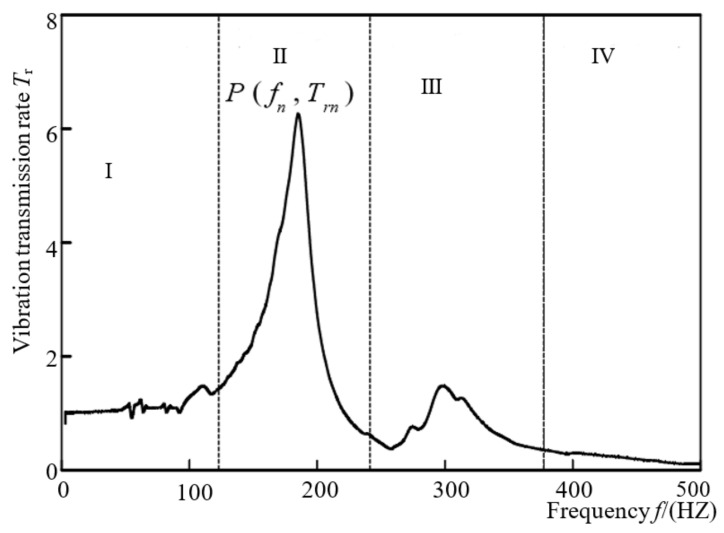
Typical vibration transmissibility curve of honeycomb paperboard (Ⅰ: Plateau region; Ⅱ: Amplifier region; Ⅲ: Fluctuation region; Ⅳ: Attenuation region) [22].

**Figure 6 materials-16-06554-f006:**
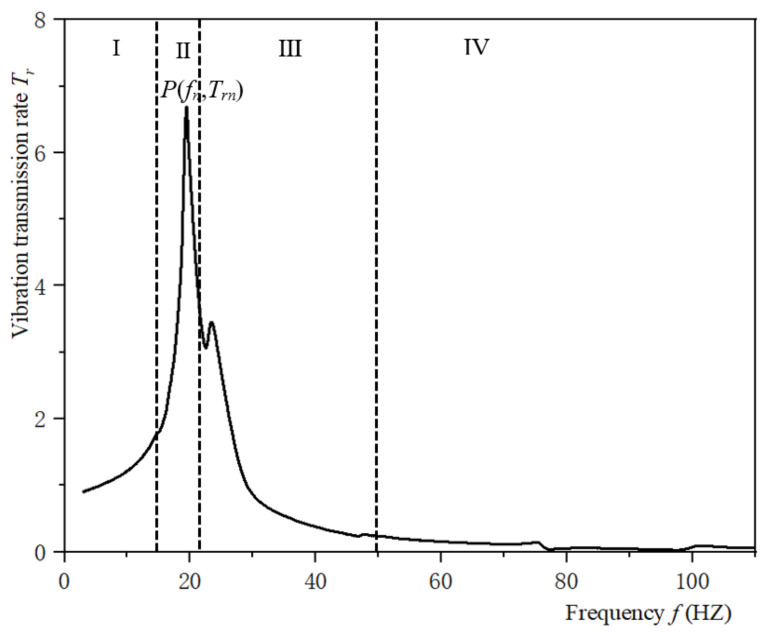
Typical vibration transmission curve of the assembly (Ⅰ: Plateau region; Ⅱ: Amplifier region; Ⅲ: Fluctuation region; Ⅳ: Attenuation region).

**Figure 7 materials-16-06554-f007:**
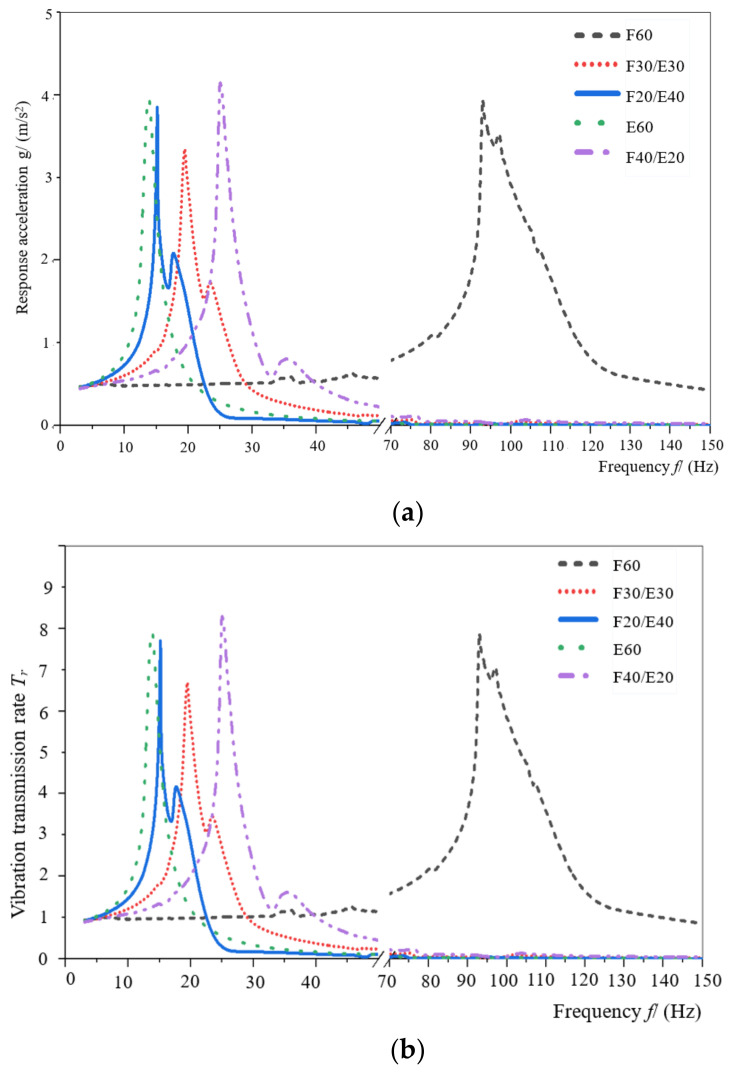
The response acceleration–frequency curves and vibration–transmission rate curves of the specimen with thickness of 60 mm. (**a**) The response acceleration—frequency curves; (**b**) vibration transmission rate–frequency curves.

**Figure 8 materials-16-06554-f008:**
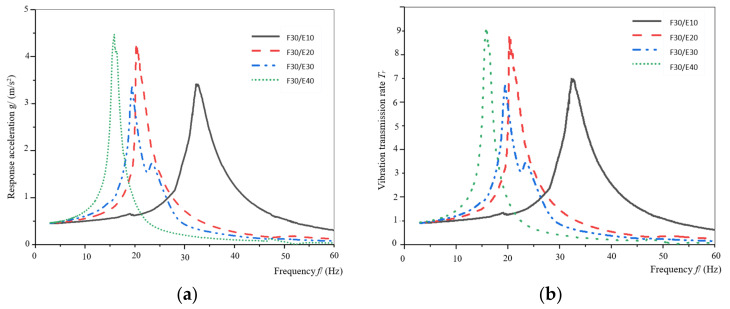
Curves of the assemblies for honeycomb paperboard with a thickness of 30 mm. (**a**) The response acceleration—frequency curves; (**b**) vibration transmission rate–frequency curves).

**Figure 9 materials-16-06554-f009:**
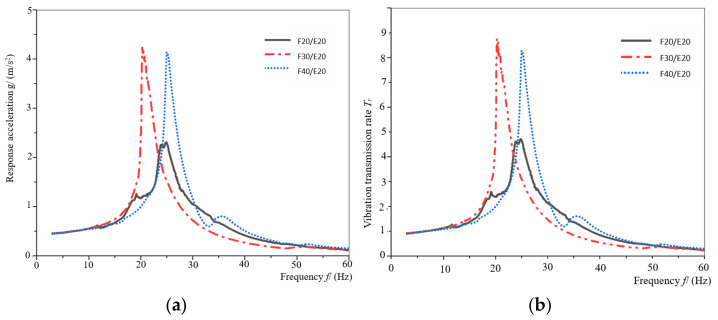
The response acceleration curves and vibration transmission rate curves of the specimen for EPE with a thickness of 20 mm. (**a**) The response acceleration—frequency curves; (**b**) vibration transmission rate–frequency curves.

**Figure 10 materials-16-06554-f010:**
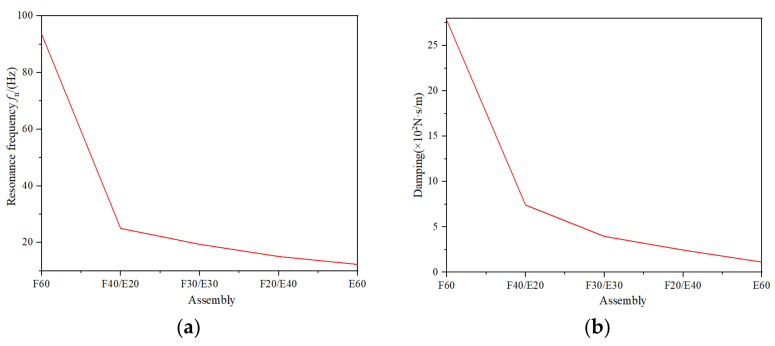
The resonance frequency and damping of the assemblies. (**a**) Resonance frequency; (**b**) damping.

**Table 1 materials-16-06554-t001:** Test specimen parameters.

Specimen	Specimen Size (mm × mm)	Thickness of Honeycomb Paperboard (mm)	Thickness of EPE (mm)	Aperture of Honeycomb Paperboard (mm)	Density of EPE (kg/m^3^)
F60	100 × 100	60		6	
E60	100 × 100		60		15.3
F30/E30	100 × 100	30	30	6	15.3
F20/E20	100 × 100	20	20	6	15.3
F20/E40	100 × 100	20	40	6	15.3
F40/E20	100 × 100	40	20	6	15.3

**Table 2 materials-16-06554-t002:** Parameters of sinusoidal sweep vibration test.

Sweep Frequency Speed (oct/min)	Sweep Frequency Range (HZ)	Target Spectral Acceleration Peak (m/s^2^)	Frequency Sweep Mode	Control Sensor Sensitivity (V/g)	Monitoring Sensor Sensitivity (V/g)
0.5 oct/min	3–150 Hz	0.5 g	logarithm	24.50 V/g	24.40 V/g

**Table 3 materials-16-06554-t003:** The damping of the assembly and the damping energy dissipation of the system for the resonance point.

Specimen	Resonance Frequency *f*_n_/(Hz)	Maximum Vibration Transmission Rate *T_r_*	Damping Ratio *ξ*	Damping *c*/(N·s/m)	*W_c_*/(J)
F60	93.6	7.89	0.06	2776.4	0.06
F40/E20	25.0	8.29	0.07	741.8	0.99
F30/E30	19.4	7.00	0.07	396.7	1.75
F20/E40	15.1	7.84	0.06	245.9	1.82
E60	12.3	6.34	0.08	113.4	0.75
F20/E30	18.1	8.58	0.06	231.8	1.36
F30/E20	20.2	8.74	0.06	302.1	1.14
F30/E40	15.8	9.17	0.05	490.4	1.75
F20/E20	24.7	4.72	0.09	217.7	0.88

## Data Availability

Not applicable.

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
