# Peer review of "An Analysis of the Vibration Transmission Properties of Assemblies Using Honeycomb Paperboard and Expanded Polyethylene"

_materials, 2023, doi:10.3390/ma16196554_

Round 1

Reviewer 1 Report

The authors have conducted interesting research on vibration transmission characteristics of honeycomb paperboard and expanded polyethylene (EPE) assembly. However, the following points must be addressed:

1.     A list of nomenclature should be included before the introduction section.

2.     Line1-2: Modify the title with incorporating application

3.     Briefly  add the significance of topic, application in the start of abstract.

4.     Line11: In the abstract it is mentioned “10 different thickness of honeycomb paperboard and EPE are obtained and compared” but not clearly mentioned that  which thickness of assembly was found the best among them ? .needed to mention the which thickness, rather than writing general

5.     Line12: taking  or assuming ? “Taking the assembly and mass block as a single degree of freedom with small damping linear system”

6.     Line 15: Write complete for the first time rather that abbreviation as “F60” What is F60 ?

7.     Line:110: Sample Identification scheme needs to be tabulated separately

8.      Line:126: Please specify the reason for choosing such environmental conditions ?

9.     ample Identification scheme needs to be tabulated separately

10.   Justify the basis of  vibration conditions and criteria , literature support is required.

11.    Result section need to revised with more literature support of similar compositions and real senecios

12.  Can we analyses how the low or high temperature effects the damping performance of these materials

13.   The authors claimed that they made various thickness. How they correlate the experimental results with the analytical models ?

14.  How much improving was made from the previously reported similar study

15.  Conclusion section needs to be revisited  as exclude “influence of these structural parameters on the vibration transmission characteristics of the assembly will be studied later. This will provide a theoretical basis for the thickness selection of  honeycomb paperboard and EPE the cushioning packaging design”

16.  Conclusions section should highlight the novelty of this research

Minor editing required

Author Response

Dear reviewer,

We are very pleased to learn from your letter about the revision of my manuscript (Analysis on vibration transmission characteristics of honeycomb paperboard and expanded polyethylene assembly. Manuscript ID: materials-2623515). Thanks for your attention and your helpful comments and advice. I have revised the manuscript according to the comments from the reviewers and corrections or modifications have been indicated by red highlighting on the main text of the paper. Please see the attachment.

Reviewer 2 Report

In the Reviewer's opinion some changes are needed in order to make the paper more valuable. Revision must be carried out according to the comments listed as follow:

1.      The introduction mentioned the impact of vibration transfer characteristics on packaging systems and products, especially when the natural vibration frequency is close to or equal to the natural vibration frequency of the product or critical part. Please provide some practical examples or examples to illustrate more specifically the potential impact of vibration transfer characteristics on packaging systems and products.

2.      Add some information on the physical and mechanical properties of these two materials to better understand their role in packaging systems.

3.      The introduction mentions the work of other scholars but does not explicitly mention the uniqueness or innovation of this study. May I ask what makes this study unique in this area and how it differentiates itself from existing studies?

4.      Does the location and mounting of these sensors affect the test results? If so, were any calibrations or calibrations performed to take these factors into account?

5.      The introduction mentions vibration transfer rate-frequency (Tr-f) curves, but there are no details in the methods section on how to calculate and plot these curves.

6.      The results section mentions that the vibration transfer characteristic curve is divided into four stages, but the frequency range of these stages is not described in detail.

7.      The results 3.1 section mentions the resonant frequency ranges of different assemblies but does not explain how important these frequency ranges are for the design of packaging systems and products. How do these resonant frequency ranges affect packaging system performance and product safety?

8.      Provide more information on how to quantify and evaluate the protective capabilities of an assembly and how this relates to vibration transmissibility?

9.      Please explain further why increasing the thickness of honeycomb paperboard and reducing the thickness of EPE will lead to an improvement in damping performance? What practical implications does this improvement have for the design of packaging systems and the protection of products?

10.   Has the study discussed how to optimize the structural parameters of the assembly to improve damping energy dissipation to meet different vibration protection needs? In practical applications, how to choose the appropriate assembly structure and parameters based on the characteristics of the product and the transportation environment?

11.   Discussion section was too brief and did not provide new information

12.Add limitations of this study and potential limitations regarding this study and possible future research directions

Moderate editing of English language required

Author Response

(The authors gave the same response as above.)

Reviewer 3 Report

SUMMARY

The article submitted for review is relevant. It analyzes the vibration transmission characteristics of honeycomb paperboard and expanded polyethylene assembly. The authors conducted an experimental refinement of the vibration transmission characteristics of honeycomb paperboard and expanded polyethylene assembly. The relevance of the study is due to the relevance of the use of such materials. Therefore, the study and various engineering and scientific approaches to obtain new knowledge about them will be welcomed by scientists and engineers. The research carried out is quite extensive. The authors varied a large number of factors in order to obtain a number of important results, which made it possible to create an empirical basis for future researchers in this area, as well as applied data for practical application. The reviewer believes that the research was carried out at a fairly high level, the interest and relevance of this article is beyond doubt. At the same time, there are a number of serious comments, so the reviewer recommends that the authors correct these comments and send the article for the second stage of review. The contents of these comments are presented below.

COMMENTS

1. The scientific task of the authors is not entirely clear. You need to understand: did the authors pursue the fundamental goal of obtaining a database of empirical data on the vibration transmission characteristics of honeycomb paperboard and expanded polyethylene assembly, or were they solving a real applied problem? There is no formulation of the scientific problem. Authors should add it to the beginning of the abstract.

2. The goal should be adjusted in terms of the nature of the research: it is applied or fundamental.

3. The methodology is described in great detail. Perhaps this should not have been done, because the abstract in this way does not reflect the main content of the article.

4. An important quantitative aspect is not disclosed by the authors at the end of the abstract. The authors say that assembling honeycomb board and expanded polyethylene of different thicknesses as shock-absorbing materials can significantly increase vibration energy absorption in logistics processes. But the authors must reflect the quantitative characteristics of this increase. The authors also talk about reducing damage to internal products. The same should be reflected in the quantitative characteristics of such a decrease. Otherwise, the practical benefits of the study are not clear from the abstract. Authors are recommended to revise the abstract.

5. Keywords do not fully reflect the essence of the article. Some of them reflect commonly known terms. I would like to see keywords in a more detailed and close search field in relation to this article and this topic. I would like to wish the authors to reconsider the keywords.

6. The authors conducted a literature review in the Introduction section, but, unfortunately, it does not fully reflect the current state of the issue. Authors must clearly reflect the transition from a review of references to their analysis, and subsequently to the formulation of a specific scientific deficit and scientific problem. At the end of the section, the purpose and tasks of the study should be formulated.

7. The authors conducted fairly large-scale studies, but did not present a program of experimental research. It is proposed to add the program in the form of a block diagram to section 2 “Materials and Methods” before paragraph 2.1.

8. Unfortunately, Figure 2 is interesting, but difficult to read, as it is made in low quality. The reviewer recommends presenting it in better quality.

9. In general, subsection 2.3 is presented without an analytical text component as a set of formulas. I would still like to see smoother text transitions between sections. The article should not only have a scientific engineering protocol appearance, but also be easily understood by interested readers.

10. The “Results” section is quite rich and contains a lot of interesting new data. The authors are encouraged to reflect and analyze these data in more detail.

11. Unfortunately, there are unreadable characters in Figures 9 and 10. They need to be presented in good quality. In addition, the chosen format of, for example, Figure 10 b is not entirely clear. A straight line between two points looks very uninformative. Perhaps the author should reconsider the design of the figure.

12. The “Discussion” section should be redone, since it does not reflect a comparison of the results obtained with the results of previous studies, including those by other authors. The authors discussed only the results presented in the figures, but this is not enough.

13. The conclusions should be supplemented in terms of the formulation of scientific novelty, prospects for the development of research and applied recommendations for the practical industry.

14. A list of 28 references for such a topic is clearly insufficient. It is recommended to supplement the literature analysis with another 10-15 works on the research topic.

Checking the language will not be superfluous.

Author Response

(The authors gave the same response as above.)

Round 2

Reviewer 3 Report

The authors have thoroughly revised the manuscript, correcting most of the reviewer's comments. In this form, the article can be accepted for publication.